# Nitrification in Premise Plumbing: A Review

**Tyler C. Bradley** [ID], **Charles N. Haas and Christopher M. Sales** *[ID]

Department of Civil, Architectural, and Environmental Engineering Drexel University,
Philadelphia, PA 19104, USA; tcb85@drexel.edu (T.C.B.); haas@drexel.edu (C.N.H.)
* Correspondence: chris.sales@drexel.edu

**Abstract:** Nitrification is a major issue that utilities must address if they utilize chloramines as a secondary disinfectant. Nitrification is the oxidation of free ammonia to nitrite which is then further oxidized to nitrate. Free ammonia is found in drinking water systems as a result of overfeeding at the water treatment plant (WTP) or as a result of the decomposition of monochloramine. Premise plumbing systems (i.e., the plumbing systems within buildings and homes) are characterized by irregular usage patterns, high water age, high temperature, and high surface-to-volume ratios. These characteristics create ideal conditions for increased chloramine decay, bacterial growth, and nitrification. This review discusses factors within premise plumbing that are likely to influence nitrification, and vice versa. Factors influencing, or influenced by, nitrification include the rate at which chloramine residual decays, microbial regrowth, corrosion of pipe materials, and water conservation practices. From a regulatory standpoint, the greatest impact of nitrification within premise plumbing is likely to be a result of increased lead levels during Lead and Copper Rule (LCR) sampling. Other drinking water regulations related to nitrifying parameters are monitored in a manner to reduce premise plumbing impacts. One way to potentially control nitrification in premise plumbing systems is through the development of building management plans.

**Keywords:** nitrification; premise plumbing; drinking water quality

---

## 1. Introduction

Water utilities across North America, Europe, and Australia have adopted the use of chloramine as a secondary disinfectant, an alternative to free chlorine, to avoid the formation of disinfectant by-products (DBPs) and to maintain a greater residual stability in their drinking water distribution systems (DWDSs) [1–4]. Monochloramine is less reactive with natural organic matter in the water, resulting in lower concentrations of DBPs [5]. Chloramine has been shown to be effective at controlling the growth of indicator organisms and reducing heterotrophic plate counts (HPC) bacteria in DWDS [1,2,6]. Monochloramine has also been shown to reduce the colonization of *Legionella spp.* in premise plumbing systems [7,8].

The use of chloramine as a secondary disinfectant has been employed since the early 1900s in the United States [9]. Surveys conducted in 2004 and 2017 by the American Water Works Association showed that the percentage of utilities using chloramine as a disinfectant, 29% and 27%, respectively, has remained stable over the last two decades [10,11]. In the 2017 survey, 188 utilities stated that they were not considering a switch to chloramine, with 34.5% of these utilities stating that a consideration for not switching was a concern about nitrification. The 2017 survey, along with a 1996 phone survey, indicated that, of the utilities that are using chloramine as a secondary disinfectant, roughly two-thirds experience nitrification to at least some degree in their DWDS [11,12].

The goal of this review is to summarize both the nitrification process in drinking water distribution systems in general and within premise plumbing systems. Premise plumbing systems are unique

water systems that can vary drastically from home to home. While other reviews exist on nitrification, this review aims to synthesize information regarding different aspects of premise plumbing systems that could impact nitrification and thus influence drinking water quality at the point of use. The research referenced within this review were found primarily using the Google Scholar database. Some references within were found using textbooks and reference manuals relevant to the field.

## 2. Background

### 2.1. Chloramine Decay

Nitrification is a biological process by which free ammonia is oxidized by microorganisms to nitrite and then further to nitrate. Free ammonia is typically added to the water at the water treatment plant (WTP) before, during, or after primary disinfection in order to form monochloramine [9,13,14]. If the correct ratios of chlorine and ammonia are added to achieve breakpoint chlorination, then water entering the distribution system may either be free of ammonia or may carry a residual. In addition to the residual free ammonia that is released from the WTP, free ammonia can be released into the system via auto-decomposition of monochloramine (Equation (1)) [15–17], monochloramine reactions with natural organic matter (NOM) (Equation (2)) [16,18], iron (Equation (3)) [19], and nitrite (Equation (4)) [17], and biologically induced cometabolism of monochloramine (Equation (5)) [18,20].

$$3NH_2Cl \rightarrow N_2 + NH_3 + 3Cl^- + 3H^+ \tag{1}$$

$$\frac{1}{10}C_5H_7O_2N + NH_2Cl + \frac{9}{10}H_2O \rightarrow \frac{4}{10}CO_2 + \frac{1}{10}HCO_3^- + \frac{11}{10}NH_4^+ + Cl^- \tag{2}$$

$$Fe^{2+} + \frac{1}{2}NH_2Cl + \frac{1}{2}H^+ \rightarrow Fe^{3+} + \frac{1}{2}NH_3 + \frac{1}{2}Cl^- \tag{3}$$

$$NH_2Cl + NO_2^- + H_2O \rightarrow NO_3^- + NH_4^+ + Cl^- \tag{4}$$

$$NH_2Cl + O_2 \rightarrow NO_2 + 2H^+ + Cl^- \tag{5}$$

The rate of auto-decomposition of monochloramine is significantly increased as the pH of the water decreases from 8 to 6 [15,17,19,21]. This increase indicates that pH plays a major role in the ability for a distribution system to maintain an effective residual throughout its DWDS. The impact of pH on monochloramine decay decreases if the Cl/N ratio is closer to 5/1 rather than 7/1 [22]. A target Cl/N ratio of 5/1 and pH of 8 is in line with ideal breakpoint chlorination targets for monochloramine formation and can reduce auto-decomposition and the formation of other undesirable chloramine products (i.e., di- and trichloramine) [15,17,23]. Temperature has a major impact on the rate of monochloramine decay [17,21]. As the temperature rises, the rate of decay increases significantly. The half-life of monochloramine decreases from 300 hours to 75 hours when the temperature increases from 4 °C to 35 °C [17].

Equation (2) makes the assumption that the chemical form of NOM in the distribution system is the same as that of typical biomass ($C_5H_7O_2N$) and that the products are as shown. Increased NOM concentrations have been shown to result in increased monochloramine decay, even at low concentrations of NOM [16,21,22]. Duirk et al. showed that monochloramine reacts directly with NOM, but the chlorine released from autodecomposition can also react with NOM in the water [22]. These reactions are important considerations as they can result in increases in the levels of DBPs in the system.

Woolschlager proposed that ammonia-oxidizing bacteria (AOB) could cometabolize monochloramine to nitrite through a chlorohydroxylamine intermediate (Equation (5)) [18,20]. He found that reaction models best followed empirical data when the cometabolism of monochloramine by AOB were included in the models. In addition to the microbiological decay of monochloramine caused by nitrifying bacteria, Sawade et al. demonstrated that microbial demands are not limited to AOB [24]. Microbial decay of monochloramine can also be caused by reactions with general biomass

and soluble microbial products (SMPs) [25,26]. The impact of AOB cometabolism of monochloramine and other bacterial monochloramine demands are discussed in more detail later in this review.

### 2.2. Microbiology of Nitrification

Whether the free ammonia is carried through the WTP or whether it is produced through decay of monochloramine, once it is not in the complexed form it is biologically available for nitrification. Nitrification is a two-step process carried out by two broad classes of autotrophic, chemolithotrophic microorganisms: AOB and Nitrite Oxidizing Bacteria (NOB) [27]. The first step of this process is the conversion of the free ammonia to nitrite by AOB (e.g., *Nitrosomonas*) and that reaction can be shown in an approximated form [28,29] or in an exact form [30]:

Approximate Equation:

$$2NH_3 + 3O_2 \rightarrow 2NO_2^- + 2H^+ + 2H_2O \tag{6}$$

Exact Equation:

$$NH_4^+ + 1.9O_2 \quad\quad\quad +0.069CO_2 + 0.0172HCO_3^-$$
$$\rightarrow 0.0172C_5H_7O_2N + 0.983NO_2^- + 0.966H_2O + 1.97H^+ \tag{7}$$

AOB must have at least two key enzymes to successfully oxidize ammonia to nitrite. The first enzyme, *ammonia monooxygenase* (AMO), is an integral membrane protein that oxidizes the ammonia to an intermediate compound, hydroxylamine. The second key enzyme, *hydroxylamine oxidoreductase* (HAO), is a periplasmic enzyme that then converts the intermediate hydroxylamine to nitrite. These two enzymes work in tandem so that the cell can successfully oxidize ammonia to nitrite and generate energy. Two of the four electrons generated from the HAO enzyme reaction are donated back to the AMO enzyme for the conversion of ammonia to hydroxylamine [31]. AOB bacteria are comprised of several genera within the *Proteobacteria* phylum, including *Nitrosomonas*, *Nitrosospira*, and *Nitrosococcus* [32,33]. These bacteria have been found in both bulk water and biofilms with *Nitrosomonas spp.* being ubiquitous throughout all of the samples taken in both pilot and full-scale chloraminated systems and *Nitrosospira spp.* found in only some samples [34–36]. Regan et al. found *Nitrosomona oligotropha* to be the most prevalent species at four utilities [35]. These bacteria have been shown to grow in biofilms within the DWDS [34,36,37].

Ammonia Oxidizing Archaea (AOA) have also been found that have the ability to oxidize ammonia to nitrite via the same mechanisms [38]. AOA have been shown to grow in drinking water systems, although there are mixed results regarding their prevalence relative to AOB [39–42]. AOA may not have the same regrowth potential in drinking water as AOB [41]. AOA are comprised of species within the archaeal phylum *Thaumarchaeota* [31], with species found in DWDS being closely related to the *Nitrosopumilus maritimus* species [39]. AOA have shown adaptations to allow growth at very low nutrient levels and are able to survive with ammonia concentrations at levels far below that of AOB [31]. This ability may result in selective pressures that cause AOA predominance over AOB in systems that experience conditions of low nutrient levels.

The second step in this process is the further oxidation of nitrite (created in the first step of nitrification) to nitrate. This reaction is carried out by NOB (e.g., *Nitrobacter*) and can be shown in both approximate and exact forms:

Approximate Equation:

$$NO_2^- + \frac{1}{2}O_2 \rightarrow NO_3^- \tag{8}$$

Exact Equation:

$$NO_2^- + 0.00875NH_4^+ + 0.035CO_2 + 0.00875HCO_3^- + 0.456O_2 + 0.00875H_2O \rightarrow 0.00875C_5H_7O_2N + NO_3^- \tag{9}$$

NOB, similar to AOB, have a particular enzyme, *nitrite oxidoreductase* (NXR), that is an integral membrane protein that allows them to oxidize the nitrite to nitrate [31]. The NXR enzyme consists of three subunits, nxrA, nxrB, and nxrC. The nxrC subunit spans the cytoplasmic membrane and the nxrA and nxrB subunits are either periplasmic or cytoplasmic enzymes depending on the species of NOB [43]. The difference in enzyme structure, either cytoplasmic or periplasmic, is one of the reasons for NOB being a diverse group of bacteria made up of seven genera within four different phyla [43]. NOB classified within the *Proteobacteria* phylum include the *Nitrobacter, Nitrotoga* and *Nitrococcus* genera [32,43]. *Nitrospira* spp. are classified within the *Nitrospirae* phylum [32,33,43]. Additional NOB species are found in the *Chloroflexi* and *Nitrospinae* phyla [43–45]. The species within the *Proteobacteria* and *Chloroflexi* phyla are structured with the *nxrAB* subunits within the cytoplasm, while the species in the remaining phyla are structured with the *nxrAB* subunits in the periplasm [43]. This structural difference in the *nxr* enzyme may result in differences in environmental adaptation, with periplasmic NOB surviving in environments with consistently low nitrite levels and cytoplasmic NOB preferring environments that have higher levels of nitrite which may occur seasonally [43]. *Nitrospira spp.* have been found to be the most prevalent NOB found in pilot and full-scale drinking water systems, with *Nitrobacter spp.* being found in some samples [34,35]. NOB have also been shown to proliferate in the biofilms of DWDS [34,36,37].

Biofilms form in DWDSs along the pipe wall through a series of stages including: attachment, extracellular polymeric substances (EPS) development, cell proliferation, maturation, and finally detachment [46]. Biofilms constitute a majority of the bacterial concentrations in DWDS and provide an environment where different bacterial species can survive cooperatively [47–50]. Biofilms have been shown to colonize quickly over a few weeks; however, in a DWDS biofilms are likely constantly changing as a result of frequent changes in hydraulic conditions and nutrient levels [49]. Simões et al. showed that biofilm growth was increased under turbulent flows and high nutrient levels when compared to laminar flows and low nutrient levels, respectively [51]. Biofilm formation allows for aerobic bacteria to increase their resilience to disinfectant residuals. This resilience is further increased with the presence of free ammonia in chloraminated systems [52]. Biofilms harbor, and facilitate the growth of, nitrifying bacteria and can promote nitrification in DWDS [36,53]. Nutrient availability has been shown to influence the community of nitrifying bacteria in the biofilm [53].

Both ammonia oxidation and nitrite oxidation reactions are aerobic and require oxygen to proceed. The majority of both AOB and NOB are obligate aerobes and require 4.57 mg $O_2$/mg $NH_4^{+-N}$ for complete nitrification from ammonia to nitrate [29]. Typically, drinking water has dissolved oxygen concentrations that can be utilized for these reactions. However, as a result, in severely nitrifying waters, oxygen can be depleted which can have negative side effects for corrosion [12]. Bodelier et al. demonstrated that, while AOB are obligate aerobes, they exhibit the ability to survive in low oxygen and anoxic environments [54]. In fact, AOB showed the ability to reduce nitrifying activities during periods of low oxygen and then to resume nitrifying activities within one hour of oxygen being reintroduced to the system. Most nitrifying bacteria have optimal growth conditions of pH between 7.5 and 8.0 and temperature between 25 and 30 °C; however, they have been known to survive in conditions that are far from optimal [55,56]. Lieu et al. found that AOB were able to survive at levels below detection limits during periods of low temperatures (typically defined as less than 15 °C), but once temperatures increased the concentration of AOB would increase again [23].

While the primary mode of nitrification in drinking water is facilitated by the aerobic AOB and NOB, it is possible that Anaerobic Ammonia Oxidizing (anammox) bacteria are present in the internal layers of mature biofilms [57]. The biofilm's matrix can act as an oxygen barrier for the internal portions of the biofilms (>50–150 μm), allowing for the growth of anaerobic bacteria within these niches [57]. Pressman et al. (2012) found that in the absence of monochloramine, the majority of aerobic activity occurred at the surface of the biofilm with very little oxygen penetration; however, once monochloramine residuals were introduced to the system, oxygen was able to fully penetrate the biofilm [52]. The ability for oxygen to penetrate the biofilm in the presence of monochloramine

indicates that anammox bacteria are only likely to be seen under severe nitrifying conditions or in the absence of initial chloramine application (i.e., disinfectant-free systems). No studies confirming the presence of anammox in DWDS biofilms could be found.

It was previously believed that nitrification was strictly a two-step process, with ammonia first being converted to nitrite by AOB and then nitrite being converted to nitrate by NOB as described above. However, Daims et al. and van Kessel et al. published concurrent studies that found novel *Nitrospira* species that, in addition to containing *nxr* genes to perform nitrite oxidation, also contained both *amo* and *hao* genes to perform ammonia oxidation to nitrite [58,59]. These species are termed Complete Ammonia Oxidizers (comammox) due to their ability to fully oxidize ammonia to nitrate. The AMO enzymes in commamox bacteria were arranged genetically in a manner different than traditional AOB AMO enzymes which made them undetectable by traditional *amoA* qPCR primers. These enzymes were located along a single gene cluster with the requisite *hao* and *nxr* enzymes, indicating they contain the genetic potential to fully oxidize ammonia to nitrate [59]. Comammox bacteria were, when isolated, able to successfully convert ammonia fully to nitrate. This species was isolated and discovered in wastewater sludges and other engineered systems, although not all *Nitrospira spp.* demonstrated the ability for comammox [58,60]. Van Kessel et al. found that their isolated comammox species were closely related (>99% nucleotide similarity) to published genomes of nitrifying bacteria in several natural and engineered systems, including drinking water distribution systems [59]. While there has been evidence of potential comammox in drinking water systems, definitive proof of these organisms in these systems has yet to be seen [42,60,61].

### 2.3. Predicting Nitrification in DWDS

In DWDS that experience seasonal changes in water temperature, nitrification events typically occur in summer months when the temperature is at its highest [12,62–66] (Figure 1). However, in DWDS in the southern United States that experience elevated temperatures all year long nitrification may be a continual issue [67]. It is not entirely clear whether it is the increased decay in monochloramine residual or the increased microbial activity of AOB/NOB that drives the start of nitrification, as both are driven by the higher temperatures typically seen in drinking water systems [17,27,55]. The combination of these two factors can lead to severe nitrification events in the summer months in areas of DWDSs that experience high water age. Once nitrification events begin, systems can experience rapid decreases in total chlorine residuals, dissolved oxygen, and pH and increases in HPC, nitrite and nitrate. Nitrifying events can result in total chlorine residual dropping rapidly from levels greater than 2 mg/L to below the detection limit within a few weeks [62,63]. As a result of these sudden and significant drops in disinfectant, there have been numerous attempts to predict nitrification before it occurs. Table 1 summarizes the different approaches, discussed below, that have been proposed over recent years.

Fleming et al. developed nitrification potential curves to attempt to identify when a water system was likely to experience nitrification [68,69]. Using a combination of Monod kinetics to model the growth of AOB and Chick–Watson kinetics to model AOB inactivation by chloramine, they developed a model to identify the conditions that will lead to nitrification within a distribution system. In this model, they make the assumption that the only limiting factors are total chlorine residual and free ammonia residual. With this assumption, they were able to develop nitrification potential curves based on total chlorine and free ammonia concentrations. They found that the lower the free ammonia concentration in the system, the lower the total chlorine residual could be before nitrification occurred. When the potential curves were implemented on full scale systems, the maximum total chlorine residual at which nitrification was expected to occur ranged from 1.6 to 2 mg/L depending on the pH of the water.

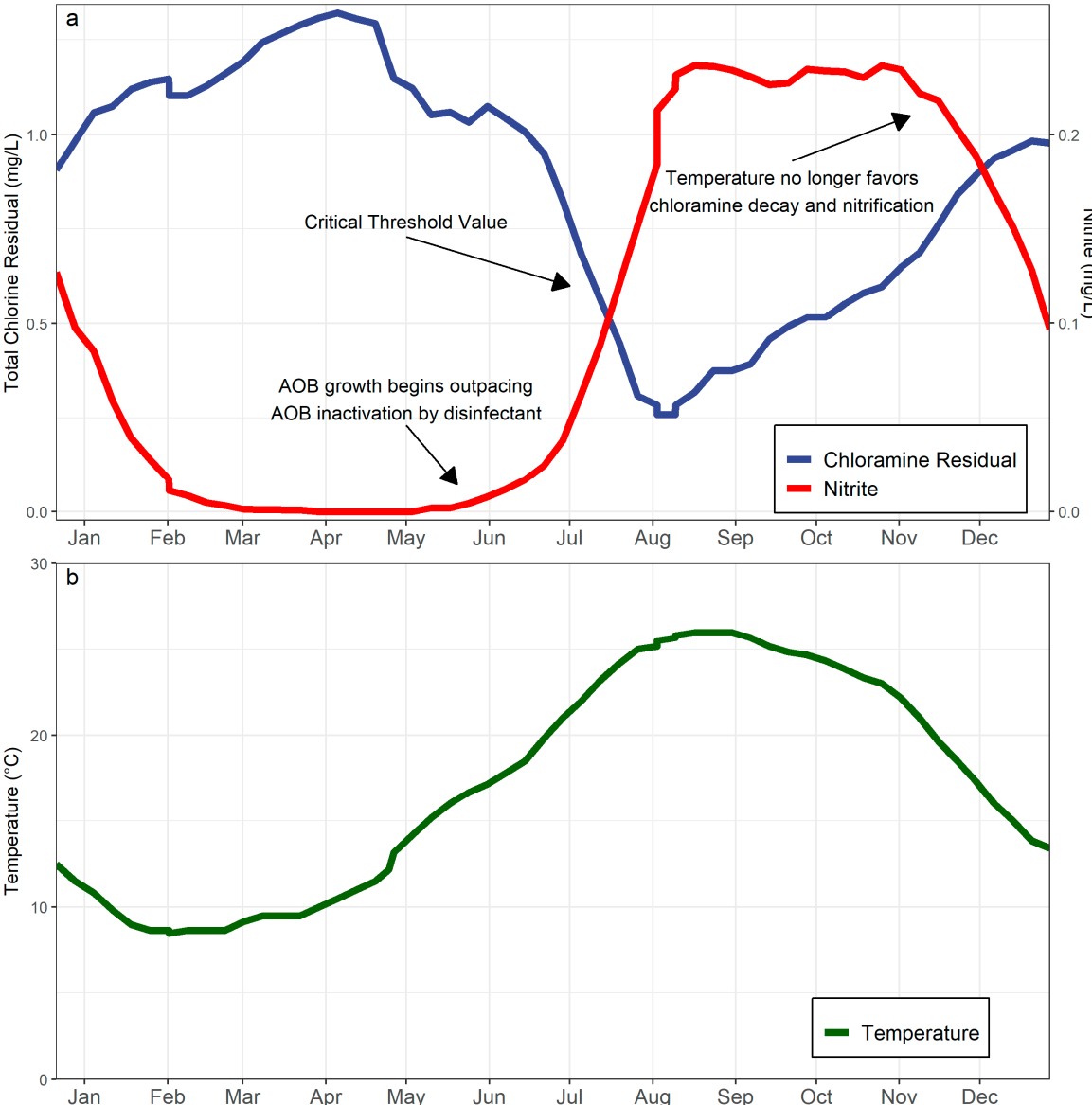

**Figure 1.** This conceptual figure illustrates what a typical nitrification trend can look like for total chlorine residual, nitrite, (**a**) and temperature (**b**) in an area of a drinking water distribution system that experiences high water age and nitrification. The point at which Ammonia Oxidizing Bacteria (AOB) begins outpacing AOB inactivation can be calculated using nitrification potential curves and is shown here as roughly the point at which nitrite is detected at the sample site [68,69] and corresponds with when temperature begins to rise above 15 °C. The critical threshold value, as defined by Pintar et al. and Sathasivan et al., is calculated using kinetic studies to identify the point at which chloramine decay increases rapidly [65,70] and has been shown to typically occur when total chlorine residual was between 0.2 and 0.65 mg/L.

Similar studies tried to identify a kinetic model that could predict the onset of severe nitrification events (i.e., situations where ammonia and total chlorine concentrations have large decreases and from which recovery would be difficult) from mildly nitrified waters, by identifying the chlorine residual, termed the critical threshold residual (CTR), that corresponds to these drops [65,70]. Mildly nitrifying waters can be defined as nitrifying water where the chlorine decay rate remains stable while severely nitrifying waters can be defined as nitrifying water where the chloramine decay begins to rapidly accelerate [70]. Pintar et al. suggested that the CTR be a set threshold for nitrite levels of 0.05 mg/L as N [65]. Sathasivan et al. suggested that the CTR should be identified by performing bench-scale decay

studies to identify the total chlorine residual at which the decay coefficient rapidly increased [70]. The bench-scale method aims to capture the total chlorine concentration at which the chlorine demand increases due to the increased AOB growth. It was shown that the CTR occurred at total chlorine residual levels between 0.2 and 0.65 mg/L.

There have been other proposals for how to best predict the onset of nitrification in distribution systems. Moradi et al. proposed a chloramine decay index (CDI) which is calculated by measuring ultraviolet absorbance at the 230 nm and 254 nm wavelengths [71]. These two measurements can be used as a ratio to determine the onset of nitrification as the 230 nm wavelength absorbance increases due to release of nitrification products (i.e., nitrite/nitrate, SMPs, or EPS) while the 254 nm absorbance decreased as a result of nitrification. Alternatively, a Nitrification Index (NI) kinetic model was proposed that accounts for the impact of ammonia and trihalomethane (THM) concentration within the water [72,73]. These models are based on the reduction in nitrification due to the cometabolism mechanisms AOB have demonstrated for different THM species that result in the inactivation of the AOB due to products of the cometabolism reaction [74–76]. While numerous methods for predicting nitrification have been proposed over the years, there is no industry wide agreement as to which approach is best for predicting the onset of nitrification. A comparison study of the different methods may prove useful to provide utilities with a better understanding of which of these proposed methods is likely to work best for them.

**Table 1.** Proposed nitrification prediction methods for drinking water distribution systems. Terms: $R_{gi}$ is the minimum total chlorine concentration needed to prevent nitrification when the concentration of free ammonia is greater than $K_s$; $K_s$ is the half saturation constant for AOB; CDI is the chloramine decay index; $A_{230\,nm}$ is the absorbance at the 230 nm wavelength; $A_{250\,nm}$ is the absorbance at the 250 nm wavelength; NI is the nitrification index; $R_g$ is AOB growth rate; and $R_i$ is the AOB inactivation rate.

| Method Name | Equation | Description | Citation |
|---|---|---|---|
| Nitrification Potential Curve | $[TotalChlorine] = \frac{R_{gi}[FreeAmmonia]}{K_s + [FreeAmmonia]}$ | Relates kinetic equations for AOB inactivation due to chloramine to AOB growth as a function of free ammonia to find chloramine residual at which growth outweighs inactivation | [68,69] |
| Critical Threshold Value | | Perform kinetic chloramine decay studies and identify the point at which decay increases by plotting decay coefficient vs middle chlorine residual | [65,70] |
| Chloramine Decay Index | $CDI = \frac{A_{230nm}}{A_{254nm}}$ | Measuring absorbance at the 230nm and 254nm wavelengths and relating the two gives the chloramine decay index. As the CDI increases, nitrification increases | [71] |
| Nitrification Index | $= \frac{R_g}{R_i}$ | Similar to Nitrification Potential Curves, relates AOB growth to AOB inactivation but includes inactivation due to cometabolism of THMs | [74–76] |

One of the major factors that impacts the onset of nitrification in a DWDS is water age. Water age can be defined as the distribution of hydraulic residence times at a given location. The hydraulic residence time at a given location at a single point in time is the amount of time it took that slug of water to reach the tap from the WTP effluent. Hydraulic residence time values are a distribution as the time it takes to reach any given point can vary depending on the hydraulic conditions in the overall system at that point in time. For the remainder of this article when discussing water age, the authors are referring to the mean hydraulic residence time from this distribution for a given location. As the water age increases, the chloramine decay increases leaving lower residuals able to inactivate the nitrifying bacteria. Several studies have shown that as water age increases, the total chlorine residual

decreases, AOB activity and concentration increases, and nitrification occurs [17,22,64,77]. Krishna et al. demonstrated that as water age increases and nitrification occurs, the microbial community begins to change as a result in both the bulk water and the biofilm [77]. The combination of *Nitrosomonas* and *Nitrospira* (AOB and NOB, respectively) accounted for as much as 15% and 9% relative abundance in the bulk water and biofilm, respectively, as the water age increased. Water age in a DWDS can extend to more than 10 days in the further reaches of large systems.

　　　An area of the DWDS that is impacted by adverse water quality related to long water age is premise plumbing. Premise plumbing is the section of the system from the corporation stop (i.e., where a home or building's plumbing connects to the water main in the street) to the customer's tap. Water can sit stagnant in premise plumbing for extended periods of time which results in long water age and can negatively impact water quality. The remainder of this review will focus on how nitrification can impact premise plumbing and what effect it can have on regulatory compliance for a water system as a result.

## 3. Nitrification in Premise Plumbing

　　　Over the last decade, the interest in premise plumbing from the water industry has increased. Numerous studies have investigated different issues surrounding premise plumbing, such as corrosion [78–81] and the growth of opportunistic pathogens [82–88]. Premise plumbing systems are an area of the distribution system that can be overlooked by water utilities, since most premise plumbing systems are not owned or maintained by the utility and most drinking water regulations do not address premise plumbing, with the exception of the Lead and Copper Rule (LCR). Premise plumbing systems are also made up of both hot and cold-water systems. The hot water systems within premise plumbing can promote increased microbial growth over cold water systems and are completely unregulated by the EPA. Premise plumbing systems can range from conventional single-family homes, multi-family residential, healthcare facilities, institutional buildings (i.e., office buildings, schools, or universities), to high-rise skyscrapers. The plumbing systems within these various building types are often unique and have varying levels of complexity (e.g., the plumbing system in a high-rise building is going to be much more complex than that of a single-family home). The varying levels of complexity in these different systems makes a one-size-fit-all solution to premise plumbing water quality issues difficult. These systems are subject to high temperatures, high surface area to volume ratios, variable flow patterns, and extended periods of stagnation [89] which can lead to increased disinfectant decay, microbial growth (including nitrification), loss of thermal control, and corrosion. These factors create a complex environment that can make it difficult for utilities to maintain high water quality in these systems. Figure 2 gives a conceptual model for how nitrification and general water chemistry conditions change as water travels throughout a drinking water distribution system and into a premise plumbing system. This section will explore different premise plumbing water quality issues and how they are impacted by nitrification and vice versa.

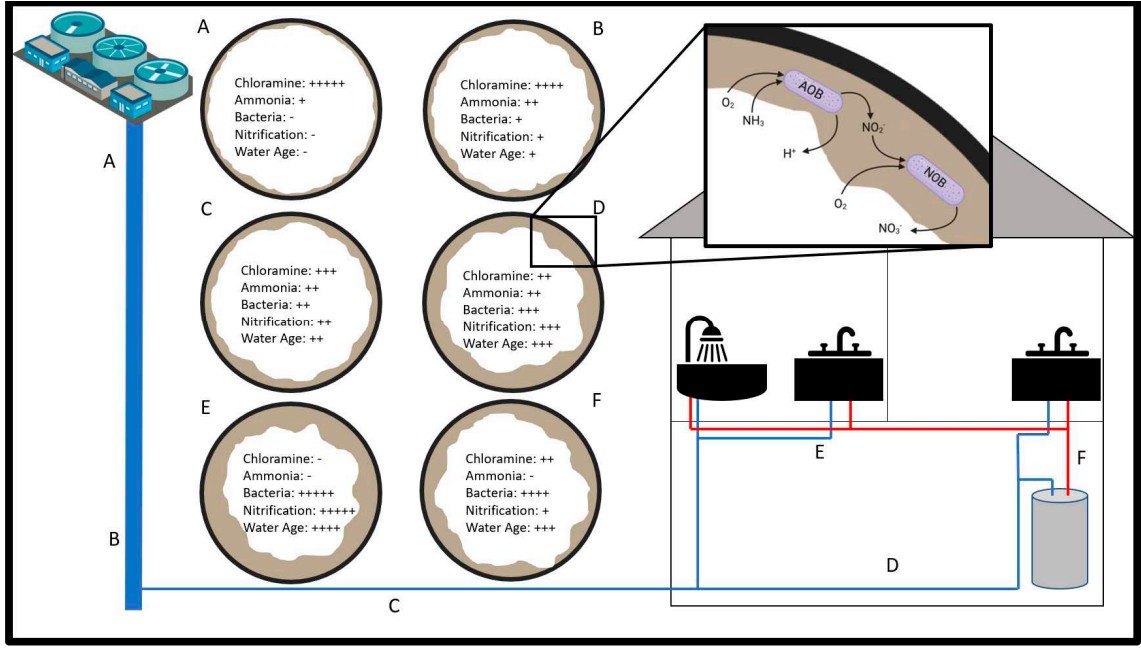

**Figure 2.** Conceptual diagram showing increases in biofilm and changes in water chemistry related to nitrification as water travels throughout a distribution system and into a premise plumbing system. (**A**) Water leaving a Water Treatment Plant has high levels of chloramine and low levels of ammonia with very little bacterial activity. (**B**) Water further into the distribution system has now had some of its chloramine decay, releasing more free ammonia into the water and increasing the bacterial activity. (**C**) Water in the service line of the home continues to see chloramine decay as water age decreases and bacterial activity and nitrification increase. (**D**) Water in the cold-water portion of the premise plumbing system is subject to moderate water ages and moderate levels of disinfectant and nitrification. (**E**) Water in distal portions of the cold-water plumbing, especially in larger buildings, is subject to very high water age, very low levels of disinfectant residual, and high levels of nitrification. (**F**) Water in the hot water portion of the premise plumbing system is subject to moderate levels of chloramine residual depending on the influent cold-water concentrations, high levels of bacterial activity and low levels of nitrification. The brown color on the inside of the pipes represents biofilm growth along the pipe wall at different stages of distribution. "−" indicates that there are negligible concentrations of a given parameter (or short residence time in the case of water age), while "+" indicate increasing concentrations of each parameter with "+" representing minimal concentrations and "+++++" representing very high concentrations relative to typical values for those parameters in a given system.

## 3.1. Chloramine Residual

Chloramine is a common disinfectant employed by utilities as an alternative to free chlorine due to the increased disinfectant stability provided by chloramine. However, in the presence of on-going nitrification, the stability of chloramine may actually turn out to be less than that of free chlorine. Zhang and Edwards demonstrated that during periods without nitrification, chloramine was more persistent than free chlorine in all pipe materials tested [90]. However, during periods of nitrification the chloramine decay rate after a six-hour stagnation was higher than that of free chlorine for all pipe materials tested. The relationship between chloramine residual and nitrification can be described as a competition between the inactivation of AOB by chloramine and the growth of AOB resulting from the utilization of free ammonia [68]. This relationship can be used to assess the biostability of a system. If the rate of inactivation of AOB in the system is higher than the growth rate of AOB, then nitrification will be inhibited under this model. However, if the growth rate of AOB is faster than the rate of inactivation than nitrification is likely to occur. The chloramine residual that corresponds to the point when growth begins to outpace inactivation has been termed the biostable residual concentration [68–70], which is

not to be confused with the critical threshold residual. Sarker et al. demonstrated that the biostable residual concentration and the relationship were dependent on temperature, with higher temperatures being associated with a higher threshold for nitrification [91]. The relationship between inactivation and growth is not unique to AOB in premise plumbing systems. Increases in biofilm growth in annular reactor studies has been shown to result in a significantly higher chloramine decay coefficient [92]. In premise plumbing systems the combination of increased stagnation times and higher temperatures are likely to result in lower chloramine residuals which reduce the rate of inactivation of bacteria in the system. These conditions will result in an imbalance in the relationship between inactivation and growth, resulting in unchecked growth of bacteria.

Prior to the research by Woolschlager et al., it was believed that the mechanisms of chloramine decay shown in Equations (1)–(3), along with increased temperatures and long stagnation times, were the primary drivers for chloramine decay in DWDS [18]. It was assumed that once nitrification was actively occurring in a system, nitrite's demand on monochloramine (Equation (4)) accounted for the increased rate of disinfectant decay. However, it has now been shown that the rate of disinfectant residual decay cannot be explained solely by chemical reactions in severely nitrifying waters [18,93]. Rather, microbiological reactions, i.e., cometabolism of monochloramine, (Equation (5)) and their products, i.e., SMPs, must be considered an important factor to chloramine decay in these systems. The co-metabolism model proposed by Woolschlager [18,20] was confirmed and shown to be performed by *Nitrosomonas europea* [94], mixed-culture nitrifiers [25], and nitrifying biofilms [26]. The cometabolism of monochloramine accounted for up to 30%–40% of decay.

Sathasivan et al. developed a method to quantify the amount of chloramine decay that can be attributed to microbiological and chemical activity, respectively, termed the microbial decay factor ($F_m$) [95]. This factor is a ratio of the monochloramine decay due to microbiological activity to the decay due to chemical reactions. While it has been shown that this method cannot be used to accurately predict nitrification events, as nitrifiers are not the only bacteria to exert a chloramine demand [24], it has been shown that the microbial decay on a system increased drastically when nitrification was on-going [24,70,96]. Sathasivan and colleagues demonstrated that the microbiological decay rate was more sensitive to increased temperature than the chemical decay rate [97]. However, it was also observed that the variability in increased decay rate due to temperature was higher for microbiological reactions rather than chemical, which was attributed to differences in responses between microbial species at different temperatures. In addition to microbial decay of monochloramine, SMPs, specifically proteins and enzymes, can exert a significant amount of demand on monochloramine [25,94,96,98]. The increased demand from SMPs could be significant in nitrifying waters, as increased microbial activity will result in higher concentrations of SMPs which can then result in further chloramine decay [25,94].

The relationship between nitrification and chloramine residual is likely to play an important role in premise plumbing systems. Since these systems are likely to experience chloramine decay due to the nature of the systems (i.e., high temperatures and stagnation), nitrification is likely to be favored. Once nitrification starts in these systems, it will further exacerbate the issue and result in further chloramine decay and the onset of severe nitrification events. The increased loss of disinfectant residual as a result of the onset of nitrification may leave these premise plumbing systems vulnerable to other water quality issues such as microbial regrowth and opportunistic pathogens.

### 3.2. Microbial Regrowth and Opportunistic Premise Plumbing Pathogens

One of the primary aims of water utilities is the delivery of water that is free of disease-causing microorganisms. Due to routine monitoring limitations for microbial constituents, indicator organisms are typically used by water utilities and regulators to identify whether they are successfully providing safe drinking water to consumers. These indicators typically include coliform bacteria (both total, thermotolerant, and *Escherichia coli*) and HPC [99–101]. Both free chlorine and chloramine are effective at inactivating biotic contaminants at the water treatment plants, however, DWDS and premise plumbing systems may be subject to microbial regrowth. Microbial regrowth is a term used to describe

the unexplained occurrence of bacteria within the distribution system [100]. While chloramine is not typically considered as strong of an oxidant as free chlorine, it has been shown to provide sufficient biocidal effects to effectively provide water free of coliform bacteria in WTP effluents [1,102]. While systems utilizing chloramine have been shown to have a more stable residual than free chlorine and lower levels of HPC and coliform regrowth [1], they are still subjective to regrowth if the disinfectant residual cannot be maintained in the distribution system [103–105].

While coliform bacteria and HPC are more traditional bacteria monitored in DWDS, opportunistic premise plumbing pathogens (OPPP) have been the focus of numerous studies in recent years due to the increasing prevalence of hospitalizations due to infections caused by OPPP. Between 1991 and 2006, there were 7,933 cases of Legionnaires' disease (only diagnosed from 1997–2006) reported; however, this number continues to rise each year with the number of reported cases reaching 6,079 in 2015 and 10,000 in 2018 alone [87,106,107]. In the same 1991–2006 study, Naumova et al. found that there were 544,643 hospitalizations from infections caused by *Pseudomonas*, and 15,861 hospitalizations due to unidentified gram-negative anaerobic bacteria in records for patients 65 years and older enrolled in Medicaid or Medicare [87]. They also found that 48,854 hospitalizations diagnosed as diseases related to non-tuberculosis Mycobacterium (NTM). It is not only the number of cases that were diagnosed in this period, but also the increasing rate at which cases related to these bacteria were being diagnosed [85]. These bacteria are all waterborne pathogens that have been associated with drinking water and premise plumbing [84,85]. OPPP typically grow in the biofilms of premise plumbing and have shown increased disinfectant resistance in drinking water systems. The introduction of chloramine has a significant impact in reducing *L. pneumophila* and the inhibition is greater over time such that a continued residual would likely be more effective at controlling *Legionella spp.* in premise plumbing systems [108]. However, the opposite was seen for NTM species, with no significant effect being seen in biofilms after chloramine introduction and a significant increase in NTM concentrations in the bulk water.

The common feature between both conventional indicator bacteria and OPPP is that they proliferate at a much greater rate in the absence of disinfectant residual. As discussed in the previous section, nitrification can result in a significant decrease in the chloramine residual. Increases in total bacteria and several OPPP were shown to be negatively correlated with chloramine residual in a simulated premise plumbing system, with increased chloramine decay being a result of nitrification [109]. The increased chloramine decay from nitrification resulted in Wang et al. finding increased concentrations of bacteria and OPPP when compared with a similarly configured chlorinated system [109]. Gomez-Alvarez et al. (2016) found that the onset of severe nitrification resulted in increases in total biomass in the system [110]. The increase in biomass and concentrations of bacteria during nitrification are likely partially explained by the assimible organic carbon (AOC) introduced to the system as a result of nitrification [111–113]. Increases in AOC along, however, have not been shown to cause increased concentrations of OPPP, indicating the importance of other nitrification side-effects on the increased growth of OPPP seen during nitrification events [109,112,113]. Given the lack of inactivation of NTM after the introduction of chloramines, reintroduction of chloramines into premise plumbing systems after periods of stagnation and minimal chloramine residual, that promoted growth of NTM, may not be able to control the growth and release of these pathogens [108].

### 3.3. Corrosion

One of the features of premise plumbing systems that can be consequential to the nitrification process is the pipe material that is used and the treatment processes that systems employ in order to prevent corrosion of these pipe materials. Plumbing materials can vary both within and between homes, with most plumbing fixes being done piece-meal, it is not uncommon for homeowners to have multiple piping materials within their homes. These materials can include copper, lead, galvanized iron, brass, and cross-linked polyethylene (PEX) and can have varying impacts on nitrification and vice versa (Figure 3).

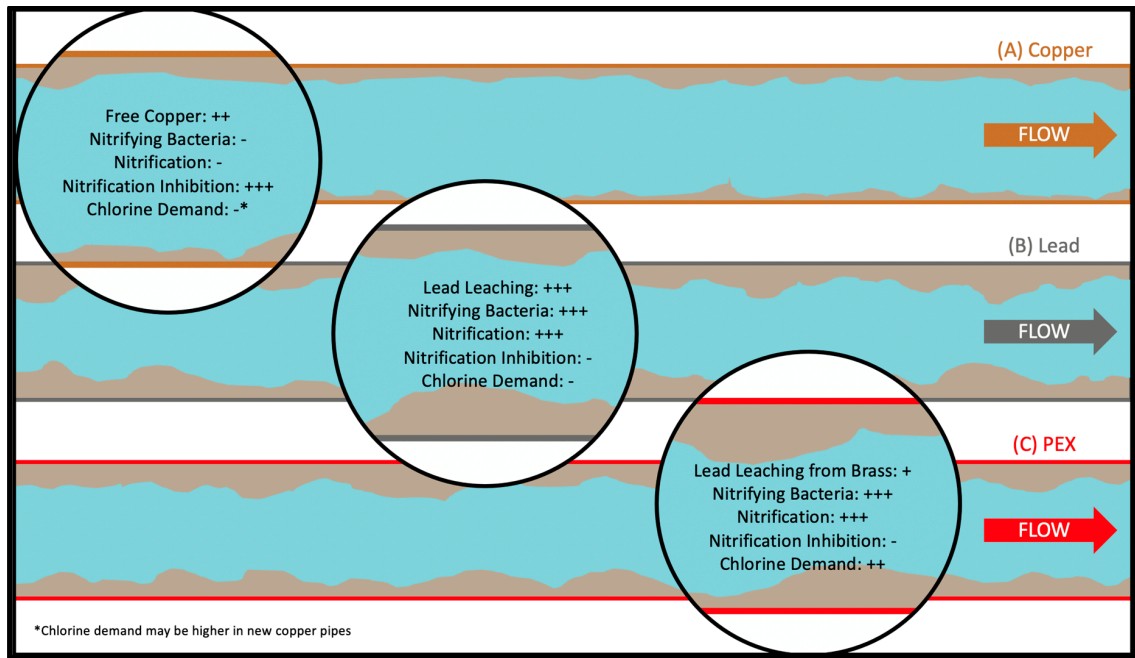

**Figure 3.** Plumbing Material can play an important role in nitrification and other bacterial activity in drinking water. In copper plumbing systems (**A**) there is limited nitrifying activity due to the toxicity of copper on nitrifying bacteria at typical drinking water concentrations. In lead plumbing systems (**B**) there is little inhibition of nitrification. This can result in drops in pH, which subsequently can cause increased levels of lead corrosion from the lead pipes. In new homes, cross-linked polyethylene (PEX) (**C**) is a popular plumbing choice. PEX has not been shown to have any inhibitory effect on nitrification, which can result in large drops in pH. This could potentially result in increased lead leaching from brass fixtures. The brown color on the inside of the pipes represents biofilm growth along the pipe wall at different stages of distribution. "−" indicates that there are negligible concentrations of a given parameter (or short residence time in the case of water age), while "+" indicate increasing concentrations of each parameter with "+" representing minimal concentrations and "+++++" representing very high concentrations relative to typical values for those parameters in a given system.

The most common premise plumbing pipe material in the United States is copper [111]. While copper is a required nutrient for the growth of nitrifiers, it is known to have an inhibitory effect on nitrifying bacteria and nitrification at concentrations above 0.05 mg/L in pure cultures [114–116]. Other studies have shown similar inhibitory impacts of copper on nitrification in environments such as wastewater treatment [117,118] and in soil [119,120]. In premise plumbing systems, increased copper leaching has been shown to limit nitrification due to its toxicity to nitrifying bacteria [121,122]. In brass pipes, nitrification was sufficiently inhibited while copper concentrations were greater than 0.1 mg/L, but once copper pipes stabilized and copper levels in the bulk water decreased, nitrification rapidly increased.

Nitrification events in premise plumbing buildings can result in significant decreases in pH levels. These drops in pH can increase copper corrosion, thereby inhibiting any further nitrification. Zhang et al. demonstrated that decreases in pH matched expected decreases based on solubility models in several pipe materials except for copper [111]. They hypothesized that increases in pH caused by copper corrosion may offset decreases in pH due to nitrification.

The toxicity of copper is dependent on the amount of free copper in the system. Zhang and Edwards (2010) showed that the impact of increased copper levels was drastically decreased when alkalinity was high (500 mg/L) compared to low alkalinity (10 mg/L) [123]. This decrease in copper's inhibitory impact is likely caused by copper complexes being formed leaving much lower concentrations of free copper in the system [123,124]. When water at these two alkalinities was dosed with 0.5 mg/L

of copper, the high alkalinity water had a resulting free copper concentration of 0.006 mg/L compared to a free copper concentration of 0.06 mg/L when alkalinity was low.

The type of corrosion control practices by the utility can also play a big role in the amount of nitrification a system with copper plumbing can experience. If a utility is using orthophosphate to control the corrosion of pipe materials, they may successfully reduce copper leaching to a level that is no longer toxic to nitrifying bacteria. As a result of this successful corrosion control, it may increase the amount of nitrification that is experienced in the system [122]. In addition to phosphorous impact on the amount of copper corrosion in the system, increased phosphorous concentrations can promote increased nitrification during periods of stagnation, as phosphorous is a key nutrient for nitrifying bacteria [123]. Some utilities use zinc orthophosphate for corrosion control [125,126] and the combination of zinc and copper at moderate levels may provide increased stability for monochloramine and reduce nitrification levels [123,127,128].

One of the historic premise plumbing materials that is commonly in the spotlight is lead. Lead can be found in premise plumbing either as lead pipe, lead-containing solder, or in brass fixtures and pipes (~0.25%–8% lead). Lead has been shown to have no inhibitory effects on ammonia utilization and nitrification in wastewater [129,130] and there have been mixed results regarding the impact of lead on nitrification in soil [131–133]. Zhang et al. showed that lead pipes had no inhibitory effect on nitrification in premise plumbing systems [122]. When compared with polyvinylchloride (PVC), brass, and copper pipe materials, lead pipes were the first to show colonization of nitrifying bacteria and complete ammonia removal. Changes to pH and phosphorous levels had no significant impact on nitrification when in lead pipes.

While lead does not appear to inhibit or promote nitrification, nitrification can cause increased levels of lead leaching into the water [111]. This increase in lead leaching as a result of nitrification is primarily attributed to decreases in pH as a result of nitrification, which can lead to increased solubility for lead [134,135]. The impact of nitrification on lead leaching is highly dependent on the alkalinity [135]. Alkalinity concentrations of 100 mg/L resulted in no difference in soluble lead levels between waters with and without nitrification. Meanwhile, at lower alkalinity levels ($\leq$30 mg/L) there were significant increases in soluble lead release in nitrifying waters. Alkalinity did not appear to have any significant impact on particulate lead release. Zhang et al. sampled homes in multiple utilities after periods of stagnation and found that homes with the greatest decreases in pH corresponded to the homes with the highest amount of ammonia consumed and the highest lead levels [135]. In addition to the impacts on lead release due to pH decreases, increases in nitrate concentration due to complete nitrification can result in increased lead release from lead solder when coupled with copper pipes [136]. Increases in lead as a result of nitrification in these studies showed lead levels far above the Action Limit (15 ppb) established by the EPA in the LCR [137].

Another prominent material in older premise plumbing systems is galvanized iron (i.e., galvanized steel or galvanized pipe). Galvanized iron piping was a dominant plumbing material until the 1960s in the United States [14,134]. Galvanized pipes are iron pipes that are coated in zinc. The zinc coating was thought to protect the pipe from corrosion, however, these pipes are subject to large amounts of corrosion and tuberculation when constantly exposed to water [14]. The onset of nitrification in galvanized pipes was slower than lead and PEX when inoculated with nitrifying bacteria and resulted in lower levels of ammonia loss [123]. The lower levels of ammonia loss were attributed to either zinc's inhibition of nitrification or potential denitrification of nitrate to ammonia by the galvanized pipe. A pipe scale investigation of nitrification impacts on galvanized iron showed that on-going nitrification reduced the amount of zinc that was released from the galvanized pipes compared to a system without nitrification [123]. In depth experiments to determine the cause of this behavior, which is the opposite of most pipe materials, indicated that reductions in dissolved oxygen were likely the main driver in the reduction of zinc release. The impact of dissolved oxygen on zinc release was further supported by comparisons of zinc release under anaerobic and aerobic conditions, in which the former exhibited significantly less zinc release compared to the latter.

A modern pipe material popular in premise plumbing systems is plastic pipes, specifically PEX. Zhang and Edwards investigated water quality impacts of several different types of plastic pipes (PEX, PVC, and polyethylene (PE)) and observed similar water quality behaviors for these materials [90]. Plastic pipe has been shown to provide very little inhibition of nitrification once the bacteria are introduced to the system [122,124]. Changes in pH and phosphate dosages had little impact on nitrification in PVC plumbing, indicating that nitrifiers can remain active outside of optimal ranges in plastic pipe [122]. It is theorized that nitrification events in homes plumbed completely with plastic piping may be more susceptible to high lead releases from brass fixtures as a result of nitrification. The lack of inhibition of nitrification even at low pH levels could result in large decreases in pH thereby increasing the solubility of the lead components of the brass fixtures [122]. Plastic pipe materials have also demonstrated a high potential for biofilm growth over other materials [138,139]. While biofilms in copper eventually reach similar cell densities to that of plastic pipes [139], the early colonization of plastic pipes may promote increased nitrification. PEX plumbing materials have also been shown to exhibit a chlorine demand of up to 0.5 mg/L of free chlorine [140]. Increased chlorine decay as a result of interaction with PEX could increase the nitrification potential in a chloraminated system.

*3.4. Green Buildings*

In the last decade, the prominence of green, or Leadership in Energy and Environmental Design (LEED) certified, buildings has increased dramatically [141]. These buildings promote environmentally friendly practices by reducing energy use, improving air quality, and reducing water usage [141]. However, the goal of increased water conservation can have negative impacts on water quality in the drinking water systems of these buildings. Water conservation can often result in extremely high water age throughout the system due to reduced potable water use. Chloramine residuals can often be completely depleted in green buildings and sites within these buildings require more than 40 min of flushing before chloramine residuals return to the levels in the DWDS [105,142]. Water age was observed to exceed several days within these buildings, while total chlorine was completely depleted within as little as one hour in some locations within the buildings. Rhoads et al. observed mild nitrification in a chloraminated system which they identify as one of the contributing factors to severe chloramine decay [142]. Increases in cold water temperature were seen in both stagnant and flushed samples relative to influent temperature, which is likely to favor nitrification [142]. Increased OPPP and microbial activity was also seen in these buildings as a result of the lack of disinfectant residual [105,142].

## 4. Regulatory Framework

In a DWDS, there are several regulatory issues that are relevant to nitrification either directly or indirectly. Nitrite and Nitrate are both regulated under the Safe Drinking Water Act (SDWA). Both have an individual maximum contaminant level (MCL), along with a combined MCL. Nitrite cannot exceed 1 mg/L as N, nitrate cannot exceed 10 m/L as N, and the sum of the two nitrogen species cannot exceed 10 mg/L [143]. In addition to MCLs, the Environmental Protection Agency (EPA) will often establish a maximum contaminant level goal (MCLG) for contaminants. The MCLG is meant to be a health based standard that utilities should strive to achieve. Often, the MCLG will be more stringent than the MCL, however, in the case of nitrite and nitrate, the MCLG is the same as the MCL. However, this regulation only applies to the entry points of the DWDS, so increased nitrite and nitrate as a result of nitrification in areas of the DWDS (e.g., storage tanks) or premise plumbing is not going to cause any regulatory violations.

Another regulatory consideration related to nitrification is maintaining total chlorine residual above the detection limit. The Surface Water Treatment Rule requires that the disinfectant residual concentration cannot be nondetectable in more than five percent of monthly samples for two consecutive months. A non-detectable chlorine residual can vary based on the state primacy agency. For instance, the Pennsylvania Department of Environmental Protection (PaDEP) recently promulgated the Disinfection

Requirements Rule (DRR) which establishes a non-detect as any value below 0.2 mg/L; an increase from the previous non-detect value of anything less than 0.02 mg/L [144]. This rule requires that no DWDS sampling site have two non-detects in consecutive months. The redefined non-detect threshold could have significant implications for systems that utilize chloramine and experience nitrification. However, the influence of nitrification occurring within premise plumbing is expected to play a small role in the regulatory compliance. Compliance samples are typically taken after flushing the tap in order to obtain a sample that is representative of the DWDS rather than the premise plumbing. However, if the tap is not flushed sufficiently and the building is experiencing severe nitrification then it is possible that a sample could be collected that is below the detection limit.

These rules are closely linked with the Revised Total Coliform Rule (RTCR) as they require that samples be taken at the same frequency as RTCR samples. Unlike the nitrite and nitrate MCLs, these regulations require sampling from locations throughout the distribution system. As a result, these samples can be impacted by the premise plumbing of the buildings used for sampling. However, utilities often attempt to limit the impact of these premise plumbing systems by letting the taps being sampled flush for three to five minutes prior to collecting both the RTCR and disinfectant residual samples. Flushing is performed with the intention of trying to monitor the condition of the water quality in the DWDS pipes and not the premise plumbing. However, if the flushing is not performed long enough to bring in fresh water from the main, then it is very possible that nitrification in one of these sampling buildings could result in both coliform positive results and non-detects for chloramine residual.

While the aforementioned regulations and the associated sampling methods are designed to minimize the impact of premise plumbing, there is one relevant regulation that specifically is designed to look at the impact of premise plumbing on drinking water, the LCR. The LCR requires that utilities develop a sampling program that allows for first-draw 1-liter samples to be taken from customer taps [137]. LCR sampling methods are designed to measure the effectiveness of corrosion control within the system. However, as discussed, if severe nitrification is on-going in the premise plumbing of one of these sampled homes, then decreases in pH could potentially result in elevated lead levels far above the action level of 15 ppb. While a single exceedance does not result in a regulatory violation, if these nitrification episodes are common throughout a distribution system, it is possible that a system could fall out of compliance with the LCR.

While there are several regulations that are relevant to nitrification in DWDS, the majority of drinking water regulations, except for the LCR, do not typically include sampling from premise plumbing systems. However, the control of nitrification in premise plumbing may not necessarily fall entirely on utilities. For traditional homes, the impact of nitrification may be minimized, as the water quality has been shown to closely match the water quality in the system after only short flushing times [145]. In larger buildings, such as high-rises, hospitals, institutional buildings, schools, and apartments, the impact of premise plumbing is likely to be more pronounced and effective management practices may need to be employed by building managers. Building managers are encouraged to develop a water management plan that can effectively identify areas with the premise plumbing system that have an increased risk of water quality issues, specifically the growth of *Legionella* [146]. A Water Research Foundation (WRF) study recently suggested that utilities should develop communication plans in order to effectively educate homeowners and building managers on how they should be best managing their premise plumbing systems [147]. These communication plans include teaching homeowners effective flushing strategies and building managers how to make water management plans. While most references surrounding building water management plans and communication plans focus on the growth of OPPP, the impacts that nitrification can have on these systems if they are receiving chloraminated water from their water supplier should be included. Nitrification can greatly exacerbate the issues that these management plans are designed to prevent, and the onset of severe nitrification may result in the original management plan being inadequate to sustain good water quality in their buildings.

The recommended steps to developing an effective building water management plan include identifying problem areas within their system, determining standard control measurements to monitor in these problem areas, and establishing corrective actions that can help bring the control measures back within desired limits [146]. The same techniques could be used to develop a building water management plan specifically for nitrification or to integrate nitrification control into existing building water management plans. An important first step to identifying problem areas within a building is understanding the quality of the influent water from the utility. If the influent water is already experiencing nitrification than the building management plan will likely have to include areas of the system that are close to the influent water as well as distal portions of the system. Conversely, if the influent water still has a high chloramine concentration and nitrification is not on going then the building management plan will likely focus on the distal portions of the cold-water system. Building managers developing these plans should be able to measure temperature and total chlorine residual reliably in order to ensure that their systems remain in control. Standard hand-held colorimeters can be used to measure total chlorine concentrations, and some are able to measure nitrite levels directly. Establishing baseline values for temperature and total chlorine in different areas of the building will help identify problem areas. Once the building manager has identified problem areas within their system to monitor, they must decide a routine monitoring plan to ensure that these locations remain within control limits. Further research is needed to determine what control limits should be set for monitoring nitrification in premise plumbing systems to ensure that it is not ongoing. Once a monitoring schedule is selected, standard response plans can be created for how to remediate nitrification within the building. The most likely response for nitrification will be developing a flushing strategy that ensures water turnover in low usage points of the building. Further research is required to determine the effectiveness of different response strategies and their longevity. Development of these plans can help building managers ensure that they are effectively managing and mitigating nitrification in their building water system and providing consumers with the highest quality water possible.

## 5. Research Gaps

Over the course of this review, the authors have identified three major research gaps that still need to be studied regarding nitrification in premise plumbing systems. These gaps include the impact of nitrification in influent water on OPPP growth in hot water premise plumbing systems, the impact of green buildings and water conservation on nitrification and vice versa, and the development of best practices regarding the implementation of building management plans focused on controlling nitrification in premise plumbing.

The first research gap identified by the authors is the relationship between nitrification and the growth of OPPP in premise plumbing systems. While there are inherent links between the effects of nitrification and a major factor in OPPP growth (i.e., loss in disinfectant residual), there are very few studies that investigate the relationship between these two processes. One area that could be of significance is investigating the growth and proliferation of OPPP in hot water systems when the receiving waters (i.e., the water entering the building or the water in the service line prior to the water heater) is undergoing nitrification. This could result in the cold water entering the water heater having very little disinfectant residual which would provide OPPP with favorable conditions for growth.

Secondly, the impacts of green buildings and water conservation on nitrification in premise plumbing systems is an area that has yet to be explored. While Rhoads et al. acknowledged that nitrification was occurring in some of the green buildings included in their study, a dedicated investigation on the relationship between the two things has yet to be done [142]. The authors hypothesize that buildings exercising water conservation practices and receiving water using chloramines as a secondary disinfectant are likely to promote nitrification in their system. A dedicated study looking at green buildings in chloraminated systems should be performed to quantify the extent to which the combination of these factors impacts water quality and what remedial techniques can be performed to reduce their impact. In addition to studying the impacts of water conservation practices

on nitrification, it would also be of interest to study the impact of influent nitrification on water quality in these buildings. If these green buildings are receiving water that is already experiencing nitrification then they may have to increase the frequency and intensity of remedial actions that are taken in order to ensure that consumers in the buildings are receiving safe drinking water.

Finally, an area with very minimal prior research is best practices for developing building management plans designed to manage and control nitrification in premise plumbing systems. There has been research investigating and developing building management plans for controlling OPPP, but none that focus on controlling nitrification. This research should focus on how to best identify where nitrification is occurring or is most likely to occur in the building and what remedial actions are most effective at reducing levels of nitrification in these areas. Determining effective control limits, or methods that can be followed to create these limits, would be extremely useful in instructing building managers on how to determine if nitrification is occurring within their systems. This research should also investigate the longevity of various remedial actions to help building managers understand how often these actions must be taken in order to continuously maintain water quality within their building.

## 6. Conclusions

Nitrification in DWDS can cause serious issues for utilities trying to maintain water quality. Nitrification is typically seen in utilities that utilize chloramine as a secondary disinfectant in order to avoid high levels of disinfection by-products and maintain a stable residual. While chloramine is known to be more stable than free chlorine under normal operating conditions, the onset of nitrification can greatly increase the rate of chloramine decay resulting in a less stable residual. Nitrification is a two-step process where AOB oxidize free ammonia, either resulting from dosing at the WTP or released from chloramine decay, to nitrite followed by NOB that oxidize nitrite to nitrate. The nitrifying bacteria that facilitate these reactions prefer temperatures that are common in distribution systems in the summer months in most climates, resulting in higher rates of nitrification during summer months in utilities that experience seasonal temperature changes.

Premise plumbing refers to the water systems that is inside of buildings. These systems range from simple single-family homes to large complex systems supplying water to high-rise buildings. Premise plumbing has been the subject of increased interest within the water community in the last decade as researchers have identified them as areas subject to increased water quality degradation. These systems are characterized by high surface-to-volume ratios, high temperatures, stagnation, variable flow patterns, and interactions with multiple pipe materials. These factors can all compound to lead to poor water quality. The onset of nitrification in these systems is likely to exacerbate the poor water quality. While there are a number of mechanisms resulting in chloramine decay in a premise plumbing system, the onset of nitrification is likely to significantly increase the rate of decay. Accounting for the biological demand on chloramine from both nitrifying bacteria and total bacteria is an important factor when trying to determine the amount of decay likely to be seen in these systems. The increase in nitrification will result in a proliferation of AOB which have been shown to be able to cometabolize monochloramine, resulting in increased decay in the presence of severe nitrification. In addition to the bacterial demand, the nitrification process can result in significant decreases in pH which has been shown to increase the rate of chloramine decay. Premise plumbing systems have also been shown, in numerous studies, to be a hotspot for the growth of opportunistic pathogens. These pathogens survive in high temperatures and can be resistant to disinfection; however, if nitrification has depleted the majority of the chloramine residual, then these pathogens can grow and proliferate unabated. These pathogens can result in diseases from gastrointestinal illness to severe pneumonia.

While the impact of nitrification on the chloramine residual and bacterial growth is a major aspect of deteriorating water quality in premise plumbing, another very important aspect is how the materials used in premise plumbing interact with nitrification. The primary material used in home plumbing is copper, which has an inhibitory effect on nitrifying bacteria. Studies have shown that copper pipes or copper released from brass materials can successfully inhibit the onset of nitrification. However,

premise plumbing systems are rarely constructed of one uniform material. There is often intermixing of multiple materials, such as copper, galvanized iron, lead, brass, and PEX. With the exception of copper and copper containing brass, the other materials did not demonstrate any inhibitory effects on nitrifying bacteria. Decreases in pH that were seen as a result of uninhibited nitrification resulted in an increase in corrosion of lead materials which could result in a health concern if exposure to lead is increased as a result.

In the current regulatory landscape, premise plumbing systems often are not accounted for. The only exception to this is the LCR which requires utilities to establish a sampling program at customers' homes at set time intervals and test for lead and copper leaching into the water. While there are few regulatory concerns for utilities when it comes to nitrification in premise plumbing systems, it is still an area of concern for them. It is hard to control the water usage and its exposure to disinfectant decay factors once it is in premise plumbing. It has been advised that utilities work with homeowners and building managers to help educate them on both why and how to maintain high water quality in their buildings. Various governmental and health-based organizations have recommended the use of building management plans for buildings that are larger than single family homes. While these building management plans are typically aimed at preventing the growth of opportunistic pathogens, the inclusion of nitrification control within these plans will likely help to ensure that the water they are delivering to the people in their buildings is of the highest possible quality.

**Author Contributions:** T.C.B. prepared the original draft of the manuscript and C.N.H. and C.M.S. reviewed and edited the manuscript for submission. All authors have read and agreed to the published version of the manuscript.

**Funding:** This research received no external funding.

**Conflicts of Interest:** The authors declare no conflict of interest.

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
