# Peer review of "Nitrification in Premise Plumbing: A Review"

_water, doi:10.3390/w12030830_

Round 1
Reviewer 1 Report
This paper provides a critical review of nitrification in drinking water systems with focus on premise plumbing. The paper cites the main seminal references on this topic but in general the paper could use some careful copy editing. Specific examples are provided below.
The term “lead and copper rule” in the abstract should be capitalized. In the last paragraph of the abstract the authors say that “One way to potentially regulate and control nitrification…”. I agree that we want to control nitrification, but I don’t think you want to “regulate” nitrification and this word should be deleted. Lines 30-32: It has also been shown to be effective at controlling Legionella compared to free chlorine. Ammonia-oxidizing bacteria (AOB) has been defined several times throughout the paper. For example, Line77 and line 87. After the first time just use AOB. This issue has also occurred with other abbreviations (e.g., SMP). Please check through the entire paper. Line 99: should “amo” be italicized or capitalized. Line 161-163: Specify what temperature is considered “low temperature”. Define “severe” and “mild” nitrification. Sathasivan et al. 2008 says during mild nitrification bulk water nitrite concentration of < 0.010 mg-N/L, chloramine decay remains stable (0.001 to 0.006 h-1). During severely nitrification bulk water nitrite concentration of > 0.10 mg-N/L and there’s accelerated decay of chloramine. In the Figure 1 caption label the y-axis. Line 193: delete the words “in DWDS”. Table 1: Define terms in equations. For example, Rg, Ri, Ks etc. Also, put spaces between number and units in 230 nm and 254 nm. Line 275. Mention that the LCR is an exception to this. Line 291: Why is “Ammonia” captilized. Figure 2: This seems more appropriate when severe nitrification is happening. There are many instances with there’s high chloramine residual in hot water systems. Line 378. Don’t italicize “Mycobacterium” here. Line 383-385: Properly italicize the microorganisms. Line 426: Change “however” to “and”. Line 449: In more modern standard this is now 0.25% lead. Line 488: You already used PVC in the paper so why spell out here.
Reviewer 2 Report
Nitrification in Premise Plumbing: A Review is a informative overview of nitrification reactions and aspects of nitrification that can impact water quality. As a peer-reviewed review article, I found it lacking from a critical review standpoint, and felt that the focus on premises pluming as indicated in the title was small relative to the general information on nitrification. I would have also expected a critical review to highlight the various research knowledge gaps that exist for possible strategies to assess and address nitrification in premises plumbing. For instance, a conclusion of the paper is that provision for considering nitrification should be made in water management programs, but I do not feel the paper informs such decisions.
Specific Comments:
Abstract
Line 19-20: What “other” regs and monitoring are you referring to? Corrosion control (pH, phosphate) is not monitored in the distribution system in many systems with history of LCR compliance. No one looks for changes in alkalinity in the distribution system. No one monitors for the organisms themselves. Only distribution system monitoring disinfectant residuals would help with this, but the manner in which they are tracked (well flushed cold water samples) ignores (at least minimizes) the impacts of storage/water age in premises plumbing. All monitoring for regs also ignores hot water systems, which may be at increased risk of nitrification relative to cold water in improperly maintained or managed systems.
Line 21-22: Was the issue of management plans revisited? There are a lot of barriers for the risk management team to effectively identify and address nitrification (education, time, resources, control limits and responses to violations), not to mention lots of unknowns of how to remediate nitrification in buildings relative to at the distribution system level.
Introduction
Line 30-32: there are also many examples, in head to head studies, where chloramines do not reduce HPCs because their disinfecting efficiency is reduced relative to free chlorine.
Line 41-42: There exist critical reviews already on nitrification generally. This review should clearly outline premise plumbing as the novel contribution.
Background
Line 51: Be clear. You are referring to ratios of total chlorine to total ammonia; “target” is not an optimal word choice here, because if you’re dosing ammonia to achieve chloramines, you don’t “target” a ratio that would achieve breakpoint.
Line 62: 5:1 In this same section, do you also want to bring up the monochloramine formation efficiency is higher a pH>8? Making most utilities target pH>8 for both formation eff and stability…
Line 72: Check grammar on plural. “reactions are important considerations“
Paragraph line starting 67: NOM is increasing in some systems due to clean air act reductions in acid rain, due to desorption of NOM at higher pH and growth of algae. This is a macro trend you might want to bring up and mention here.
Line 155: nitrified? waters
Section on microbiology: The microbiology of nitrification is fairly established (the vast majority of the references from this section are more than a decade old). Not to say that there’s not really interesting stuff to still study, but refocusing on this section on premise plumbing (I’m writing this comment before reading the entire paper) and implications therein, might be more interesting and useful. For instance, in line 160-163, you mention that nitrifiers can be dormant in cold temperatures. For distribution systems, this means that they can start to regrow after periods of colder water; but the implication for premise plumbing is that cold water in the winter that normally doesn’t grow nitrifiers may continue to be an issue in premise pluming, particularly in buildings with high amounts of water conservation.
Figure 1: I find this figure perplexing. There are no y-axis labels. Is this real data or a conceptual graphic? How where the data collected, or, alternatively, what is the source of this information? You point out a critical threshold in here, but do not allude in the text to how that is determined or what that value represents? If it’s where nitrite = chloramine, there are two critical thresholds, but they are at different y-axis values…in the table, the critical threshold value is associated with kinetic chloramine decay profiles, but that information is lacking in the figure? Does the temperature threshold alluded to in the note on the figure agree with the theoretical values you’ve reported earlier? Can you report temperature? The point at which AOB growth>AOB death by disinfectant is also confusing, as there are lower chloramine residuals in Jan that have no issues with nitrification (due to temperature??). Does this note need another modifier about temperature then? Are we to assume that the high levels reported are similar to what the water treatment plant doses? If so, this suggests that the activity of the disinfectant is reduced at levels that are like 80% of the average highest residual during periods of no nitrification. In distribution systems, it’s not atypical to see losses of 50% or more of the total residual and not have issues with nitrification. I think there’s a lot of information missing here to fully understand what is going on with these “data”? (if they are real data)
Table 1. Can you define all variables below the table, Rg, Ks, Ri, etc.
Line 273-275: something that may be worth mentioning here, is that many PP issues with microbial growth occur in the hot water, which is 100% unregulated by EPA.
Line 276: apartment should be multifamily residential (covers a wider array than just apartments); healthcare is omitted but should be listed.
Line 277: variation in complexities is a weird way to state this; is it more the variability that makes it complex rather than variation in how it complex?
Line 281: nitrification is included in microbial growth? Missing from this list is decrease thermal control.
Figure 2: This figure also perplexes me. You should define a legend for the +/- part. For instance, for the “+” for ammonia, are you referring to total ammonia or free ammonia? It’s unclear what a “-“ ammonia means in this context. You should also define what the brown stuff is. In iron mains there would likely by much more “brown” stuff than in premise pluming due to tuberculation. Is that just nitrifying related bacteria biofilm? For E, chloramine actually survives quite well in hot water systems relative to other disinfectants (again, a legend for the +/- would be helpful). Also, a thermal shock in the hot water may inactivate nitrifiers, which are very sensitive to temperature >~30 C. The D label and E label appear to only be on the service line/supply side. There are vast differences in water quality that can take place between a cold water service line and a stagnant cold water line in the PP. Same for hot water supply – in fact, conceptually, there would be very low potential for nitrification in line you’ve labelled E, as this would be the highest use and highest temperature portion of the residential system you’ve modelled.
Section 3.1: this seems out of place to me, in at least the first paragraph. It discusses general chloramine and nitrification concepts, and – for me – doesn’t necessarily fall under the PP label. You want to separate “distribution” system level knowledge about nitrification from PP, highlighting the research gap regarding the fact that nitrification in PP is totally ignored.
Line 326: consider alternative wording. undergoing? once nitrification had been initiated? established? once nitrifiers were actively growing?
Line 323-333: I’m not seeing what this paragraph has to do with PP. Think about the knowledge, skill, and tools PP people have available to them. What do they need to know about nitrification in their system? how should the monitor for it? In the event that they are dosing a disinfectant to their system, at what point(s) should they consider changing what they are doing?
Line 363: Free chlorine and chloramine oxidize and inactivate microbes; “remove” implies a physical process
Line 374: This is an underestimate of all diseases reported. 2018 there were 10K cases reported to CDC, for instance. The underestimate / out of date reporting period should be highlighted
Line 388-399: Should you mention the AOC thresholds and how much Organic C nitrification can fix as a result of its growth. It has been suggested in past literature that this organic carbon fixed could support growth to OPPPs
Figure 3: I have a similar issue as with Figure 2. I do not understand, qualitatively or semi-quantitatively what the + and – represent. For instance, Copper gets a +++ for N inhibition (assuming because Cu(II) released from water can act as a biocide; but you have one + for PEX lead leaching even though you say that that decreases in pH can be an issues for it. I don’t know how these are on the same scale. Why is there not a chloramine decay category for the other materials? Copper would be higher than PEX in some circumstances but not others (new pipes, corrosive water; but not aged copper).
Section 3.4: The extent to which green buildings contribute to increased nitrification (and the WQ impacts), should be quantified in a controlled field study. The work by Rhoads and Edwards only acknowledged the problem, but didn’t really work to quantify or explore other aspects of it.
Line 519: Relevant, not prevalent
Paragraph starting line 519: As with the next paragraph, it is important to point out where and how those samples are collected, right? regulatory compliance is distribution system, on a running average basis or consecutive month basis. This is important to understand if people are going to be collecting samples.
Line 552-554: Can you provide a reference for this statement?
Line 635: It would be nice, to cover some of the how in this review, beyond having a building water management plan. There are lot of issues and unknowns associated with how management teams can effectively detect issues with nitrification and, if they do detect them, what they should do about them.
Reviewer 3 Report
Thank you for the opportunity to revise this interesting manuscript.
Even though it deals with a very important issue (i.e. nitrification in premise plumbing), I have some major concerns about the present form of the article.
In particular:
1) the introduction section is too short. The Authors do not mention any information about the problems which have led to the choice of using chloramine in water disinfection. The only mention to the DBPs is in the line 28, but no information is provided on the health problems related to the presence of DBPs in potable water. Furthermore, the authors only refer to United States, United Kingdom, and Australia but the DBPs' problems were recurrent in Europe, too. Several European countries are now using chloramine in order to avoid DBPs in drinking water. I strongly suggest the authors expand the introduction background, focusing attention on a wider panorama. For example, different European countries (particularly Italy) have asked for derogations to the water quality standards for years, just because the drinking water supply was affected by the presence of DPBs . The situation has led to the highest population distrust of the drinking water quality that we can see either in the USA and in Europe. For these reasons, the choice of changing the disinfectant and use chloramine was made. I can suggest some reference the authors can consider for describing this aspect:
Azara A, et al. Derogation from drinking water quality standards in Italy according to the European Directive 98/83/EC and the Legislative Decree 31/2001 - a look at the recent past. Ann Ig 2018;30:517-26. doi:10.7416/ai.2018.2252) Dettori, M.; et al. Population Distrust of Drinking Water Safety. Community Outrage Analysis, Prediction and Management. Int. J. Environ. Res. Public Health 2019, 16, 1004. Azara A, et al. First results on the use of chloramines to reduce disinfection by products in drinking water. Ig Sanita Pubbl 2010;66:583-600.
2) The main problem I had while reading the manuscript is that the authors present it as a review but it has not the structure of a scientific review (neither a systematic nor a narrative review). The manuscript sounds more like a concept paper. I strongly suggest the author consider the possibility of:
1) if the Journal admits it, modifying the article from a review into a concept paper;
2) add a paragraph to describe the selection process of the articles and the databases used for the narrative (not systematic) review.
3) The title of the manuscript must follow the consideration aforementioned...
Reviewer 4 Report
The manuscript concerns nitrification in premise plumbing: a review. The following suggestions should be referred to. The following suggestions should be referred to. The abstract should be more concise. The review discusses factors within premise plumbing that are likely to influence nitrification. A review is prepared in a good manner and consists of three figure, which is prepared by the authors of the manuscript.
Round 2
Reviewer 1 Report
The authors have sufficiently addressed by concerns.
Author Response
No comments to address.
Reviewer 3 Report
Thank you for the opportunity to revise the new version of the manuscript.
The authors have partially addressed the issues raised during the first round.
On the one hand, I agree with the authors' decision to focus the attention on the chloramination and nitrification issues instead of the presence of DBPs in drinking water. On the other, the authors have not addressed the main request for referring to a wider international panorama in the introduction, as the chloramine is used not only in the USA, UK, and Australia.
The other (more important) issue raised (and not addressed) was related to the review structure of the paper. Even though there are 140 references reported, it is important for the readers to know the selection process of these papers, in order to assure the reliability and reproducibility of the study. I suggest the authors add a short paragraph in the methods section, clarifying how did they found the articles, the databases and the keywords used for the research.
Round 3
Reviewer 3 Report
The authors have only partially addressed the requests, and the reliability and reproducibility of the study are not verifiable.
For these reasons, I regretfully cannot recommend the publication of the manuscript in Water.